# Nucleosome-Omics: A Perspective on the Epigenetic Code and 3D Genome Landscape

**DOI:** 10.3390/genes13071114

**Published:** 2022-06-22

**Authors:** Siyuan Kong, Yuhui Lu, Shuhao Tan, Rongrong Li, Yan Gao, Kui Li, Yubo Zhang

**Affiliations:** Shenzhen Branch, Guangdong Laboratory for Lingnan Modern Agriculture, Key Laboratory of Livestock and Poultry Multi-Omics of MARA, Genome Analysis Laboratory of the Ministry of Agriculture and Rural Affairs, Animal Functional Genomics Group, Agricultural Genomics Institute at Shenzhen, Chinese Academy of Agricultural Sciences, Shenzhen 518120, China; kongsiyuan@caas.cn (S.K.); yuhuilu13@163.com (Y.L.); tanshuhao9999@163.com (S.T.); lirongrong@caas.cn (R.L.); 18847950417@163.com (Y.G.); likui@caas.cn (K.L.)

**Keywords:** epigenetics, nucleosomes, MNase-seq, Micro-C, 3D genome

## Abstract

Genetic information is loaded on chromatin, which involves DNA sequence arrangement and the epigenetic landscape. The epigenetic information including DNA methylation, nucleosome positioning, histone modification, 3D chromatin conformation, and so on, has a crucial impact on gene transcriptional regulation. Out of them, nucleosomes, as basal chromatin structural units, play an important central role in epigenetic code. With the discovery of nucleosomes, various nucleosome-level technologies have been developed and applied, pushing epigenetics to a new climax. As the underlying methodology, next-generation sequencing technology has emerged and allowed scientists to understand the epigenetic landscape at a genome-wide level. Combining with NGS, nucleosome-omics (or nucleosomics) provides a fresh perspective on the epigenetic code and 3D genome landscape. Here, we summarized and discussed research progress in technology development and application of nucleosome-omics. We foresee the future directions of epigenetic development at the nucleosome level.

## 1. Introduction

The nuclear genetic code of eukaryotes (such as animals and plants) depends on chromatin, which is a highly organized complex of DNA and proteins and is a principal component of the cell nucleus [1]. What is amazing is that chromatin almost carries most of the genetic information about organic development, growth, and aging. With conventional electron microscopy measurements, it was found that chromatin containing roughly 2 m DNA is condensed and folded in the nucleus, which is only about 10 μm in diameter [2]. The diameter of a classical chromatin fiber is just only ~30 nm [3]. Scientists have reached a consensus that nucleosomes could be thought of as the basic building blocks of chromatin [4]. In addition, the “beads on a string” conformation vividly depicts the state of nucleosomes in chromatin [5]. During the different biological processes, such as cell division, cell proliferation, and differentiation, the “beads on a string” nucleosomes can flexibly compress and loosen with DNA duplication and transcriptional regulation. Nucleosomes are basal structural units, which consist of histone proteins and organizers. They are assembled into compact structures called chromatin in interphase, or into enormous, high-order structures called chromosomes in mitosis [1]. Scientists have used classical physics tools and methods, electron microscopy, X-ray scattering and crystallography, etc., and found or proposed some models [3]. For example, from the 1970s–2010s, there are the one-start solenoid model [6], a two-start helical ribbon model [7], a two-start crossed-linker model [8], a polymer melt model [9], and a two-start zig-zag model with tetranucleosomal unit [10]. This makes the nucleosome folding structure and pattern more and more clear [3]. Therefore, in the early days of nucleosome research, especially twenty years ago, these studies mainly focused on ultrastructural observation and structural model analysis [3]. However, the effects of chromatin conformation and nucleosome dynamics on transcriptional regulation have been poorly studied and unclear. Recently, with the development of next-generation sequencing (NGS) technology, various nucleosome-resolution or even base-pair resolution-level genomics sequencing technologies have been developed in succession. For instance, based on nucleosome-level genome fragment digestion by micrococcal nuclease (MNase), MNase-seq and derivative technology [11], MNase-Exo-seq [12], MNase-ChIP-seq [13], MACC-seq [14], etc. was developed. With these technologies or integrating with other omics technologies (ChIP-seq, RNA-seq, or others), accurate nucleosome position, the distribution of specific genomic regions, and their impacts on transcriptional regulation and gene expression were revealed [15]. Fortunately, it is worth pointing out that the development of three-dimensional (3D) genomics technologies makes it possible to analyze the three-dimensional conformations of chromatin. Hi-C and ChIA-PET are the representative technologies that burst the field of epigenetic research [16]. By combining Hi-C with MNase genome fragmentation, Micro-C and its derivative Micro-C XL can be improved [17]. These technologies help unmask ultra-fine-scale chromatin ultrastructure at the 100–1000 base-pair resolution [18]. Some special structures, dots, and stripes, which are not easily found in the Hi-C, are well represented and reconstructed [19]. In this review, from the history of nucleosome discovery to the revelation of the chromatin hierarchy, we introduce advances in nucleosome-omics technologies and further expand on bioinformatics analysis tools and processes for nucleosomics. Finally, the review tries to depict future perspectives of nucleosome-level genomics technologies.

## 2. Nucleosome

### 2.1. Scientific Discovery, Development and Related Research of Nucleosomes

Chromatin is the carrier of genetic and epigenetic information about organisms. Under an optical microscope, at the 1400 nM length interval, we could observe some regular-shaped and compact chromosome packages called metaphase chromosomes, and sometimes, loose and seemingly irregular chromatin in other interphase cells [20]. The chromosome packages the genome via a hierarchical series of folding steps ranging from 700 nM, 300 nM, to 30 nM (Figure 1a) [6]. In addition, the chromatin can directly extend 30 nM chromatin fibrils (Figure 1a) [20]. Thirty nM chromatin fiber’s substructure and its higher-order structure are hot spots in the current spatial chromatin conformation epigenetics research [10]. Physical and biochemical approaches complement each other to study the chromatin spatial conformation [3]. By using microscopic imaging techniques, ChromEMT, the nucleus, chromatins, nucleosomes, and chromatin loops are visualized [20].

During the 1970s, the nucleosome and nucleosome “beads on a string” model were first discovered and proposed (Figure 1a) [21]. Nucleosomes are the basic structural units of eukaryotic chromatin, which is composed of DNA and histones. Under the 11 nM diameter sub-chromatin fiber, each nucleosome consists of 145–147 bp of DNA wrapped around 1.75 histone octamer loops, where the histone octamer consists of two copies of the four core histones H2A, H2B, H3, and H4. Histone isoforms H1 are a linker histone [22]. Nucleosomes control cellular processes by affecting transcription factor accessibility to DNA [23]. Aggregation and deaggregation of nucleosomes are consistent with closed chromatin (heterochromatin) and open chromatin (euchromatin) (Figure 1b), which are related to transcriptional repression and activation. When the genomic regions have little nucleosome occupancy and become nucleosome-depleted regions (NDRs), the protein factors (pre-transcriptional initiation complex, chromatin remodelers, complex enzymes, histone chaperone, histone variants, transcription factor, or RNA polymerase II will access the region and affect the transcriptional expression of nearby genes (Figure 1c) [24].

As is known, DNA methylation and histone modification are two important directions of classical epigenetic research (Figure 1a) [25,26]. They have important associations with gene function and expression patterns. The nucleosome position and histone components change are important parts of chromatin remodeling (Figure 1c) [27,28]. They are critical for gene expression and most DNA-level processes. Besides, nuclear phosphatidylinositol phosphates (PIPs) are also critical interactors and regulators of chromatin remodeling and transcription and thus are crucial for establishing a transcriptionally competent landscape of the cell nucleus [29,30]. The nucleosome position is affected by many factors, such as DNA sequence, transcription factor, RNA polymerase II, preinitiation complex, and ATP-dependent nucleosome remodeling enzymes [27]. Scientists cannot find any one factor alone that determines the nucleosome position. This is the result of their combined effect. Sometimes, the histone components are not classical octameric units; the canonical histone is replaced with a histone variant during special biological processes and altered gene expression patterns [31]. Thus, the mobilization, disassembly, and destabilization of nucleosomes are new insights that can be a breakthrough for exploring the mechanism that influences gene expression regulation and other eukaryotic DNA processes [24].

### 2.2. Significance of Studying Chromatin Nucleosome and Nucleosome-Level DNA

The packaging of eukaryotic genome into chromatin affects every process that occurs on DNA. Different cells have different nucleosome organization characteristics, and nucleosomes can occupy specific locations in some cells but not in others [33]. The positioning of nucleosomes on the underlying DNA plays a key role in the regulation of these processes [27]. The presence or absence of nucleosomes in a gene expression regulatory region determines whether transcription factors can bind to that particular region [34]. Moreover, the arrangement of nucleosomes along the DNA sequence is also a regulatory mechanism that affects gene expression and other DNA-dependent processes [35]. Therefore, to understand gene regulation quantitatively, it is necessary to know not only the precise location of nucleosomes (i.e., nucleosome positioning), but also the percentage of cells that contain nucleosomes at a given location (i.e., nucleosome occupancy) [27].

With the development of next-generation sequencing technology, it is possible to accurately locate nucleosomes at the genome-wide level and calculate the nucleosome occupancy rate of the whole genome [36].

## 3. Genomics Techniques and Progress for Studying Epigenetic Phenomena at the Nucleosome Level

### 3.1. Nucleosome-Level Genomics Technology and Related Research

Micrococcal nuclease sequencing (MNase-seq) is a sequencing technology first developed to study nucleosome localization. This technology represents the starting point for nucleosome-omics technology [36]. MNase is derived from Staphylococcus aureus and has endonuclease and exonuclease activities. According to the characteristics of the enzyme, after the chromatin DNA is fully digested, most of the naked DNA between nucleosomes is digested, and the DNA wrapped around the histone octamer is protected from being digested [37]. After DNA extraction, these DNAs are DNA fragments attached to nucleosomes. Then, DNA libraries can be constructed and sequenced by NGS technology, and the nucleosome localization map can be drawn using bioinformatics mapping (Figure 2, MNase-seq) [38].

MNase is the endonuclease of choice when isolating nucleosomes from chromatin after the digestion of linker DNA. In history, MNase is widely used for the localization of nucleosomes [39]. Once double-strand DNA breaks are introduced, nucleases digest the exposed ends until they encounter obstacles, such as nucleosomes or other proteins that bind to DNA [12]. Tested by titration under concentration or time gradients [11,15], the isolated DNA exhibits a ladder effect when resolved by gel electrophoresis, with each step of the ladder corresponding to nucleosome-protected DNA. The smallest rung is usually around 150 bp (corresponding to mononucleosomal DNA), followed by 300 bp (dual-nucleosomal DNA), 450 bp (tri-nucleosomal DNA), etc. [15]. Mononucleosomal DNA is then extracted from the agarose gel and processed accordingly. Therefore, based on the above principles, MNase-seq derivative technologies were developed. Table 1 summarizes the different MNase-derivative experimental techniques for nucleosomics. It includes single-cell MNase-seq (scMNase-seq) [40], Ultra low-input MNase-seq (ULI-MNase-seq) [41], MNase-Exo-seq [12], ChIP-MNase [13], Nucleosome occupancy and methylome sequencing (NOMe-seq) [42], MNase accessibility sequencing (MACC-seq) [14], MNase hypersensitive sites sequencing (MH-seq), and Array-seq [43,44]. The cell input of the starting material, enzyme digestion, sequencing type, advantages, and shortcomings and features are shown and discussed in Table 1.

MNase-seq and derivatives help scientists discover many phenomena that cannot be obtained by traditional optical microscopy. For instance, Ramachandran et al. [45] revealed subnucleosomal protection mechanisms in Drosophila cells using MNase-seq technology. At the first nucleosome downstream of the transcription start site (+1 position), unencapsulated intermediates were found, including hexamers lacking proximal or distal contacts. Simultaneously, they found that unpacking was increased by inhibiting topoisomerase or depleting histone chaperones, while inhibiting the pausing of RNAPII release or reducing RNAPII elongation decreased unpacking. Further investigation revealed that the positive torsion produced by prolonged RNAPII resulted in a transient loss of histone-DNA contacts. By analogy, they found that nucleosomes flanking the insulating site of human CTCF were similarly disrupted [45]. Furthermore, we illustrate some kinds of scientific discoveries have been made using these MNase-seq derivate technologies. Single-cell micrococcal nuclease sequencing (scMNase-seq) was explored to detect genome-wide nucleosome orientation and chromatin accessibility from single cells or a small number of cells by Gao et al. [40] Compared with MNase-seq, the advantages of scMNase-seq is that it can become a single-cell suspension from dissociated tissues. It is significant to reveal epigenetic heterogeneity during the period of normal tissue development or pathological processes and reveal important information for highlighting the molecular mechanisms [40]. To illustrate the new mechanisms of chromatin remodeling, Wang et al. [41] used ultra low-input MNase-seq to analyze the dynamic changes of genome-wide nucleosome occupancy and positioning of mouse embryos during the first 12 h after fertilization. They revealed some novel molecular regulation mechanisms of the ZGA process, including the detailed pattern of NDR rebuilding in promoters and nucleosome positioning dynamics in TF-binding motifs in mice [41]. Based on MNase-seq, Ocampo et al. [12] introduced E. coli exonuclease III (Exo III) and MNase in simultaneous digestion of chromatin and reconstituted nucleosomes in budding yeast nuclei. They discovered two new intermediate particles that formed when it removes histone H1. They showed the difference among core particles, proto-chromatosomes, and chromatosomes (Figure 2, MNase-Exo-seq) [12]. By analyzing the histone content of all MNase-sensitive complexes using MNase-ChIP-seq, researchers found that yeast promoters are predominantly bound by non-histone protein complexes and MNase sensitivity does not indicate instability, but rather the preference of MNase for A/T-rich DNA compared with G/C-rich nucleosomes [46]. Since MNase could recognize more open chromatin regions than DNase and Tn5, Zhao et al. [43] developed a technique called MNase hypersensitivity sequencing (MH-seq), which was used to show the open chromatin regions in Arabidopsis thaliana. Compared with DNase-seq or ATAC-seq, the genome of open chromatin regions identified by this method has a wider range. This provides a new method for the recognition and classification of open chromatin in plants and animals [43]. Array-seq is another technology, including some steps of electrophoresing the DNA digested by MNase on the Bioanalyzer equipment, and selecting the samples with the appropriate degree of enzyme digestion to sequence on the Oxford Nanopore Minion sequencer. Using this method, Baldi et al. [44] determined array regularity and linker length throughout the Drosophila genome, even in non-localizable regions, and revealed an inverse correlation of regularity with transcriptional activity [44]. ChIP-exo is the high-precision version of ChIP-seq. It was applied to map the accurate DNA fragments bound by specific proteins in mammalian cell lines. ChIP-exo was used to reveal the genome-wide binding sites of GATA1 and TAL1 transcription factors in mouse erythroid cells. They found that TAL1 is directly recruited to DNA by protein–protein interactions with GATA1 during erythroid differentiation [47,48]. To understand the dynamic structural properties of chromatin, the technology requires the ability to analyze open and closed chromatin regions, and meanwhile, to assess nucleosome occupancy. MACC technology allows genome-wide simultaneous measurement of chromatin opening, compaction, and nucleosome occupancy in a single assay. MACC interrogates each genomic locus to measure nucleosome location and accessibility. MACC can also perform histone immunoprecipitation steps to observe the protective mechanism of histones [14]. Traditional MACC-seq requires a large number of cells, and Lion et al. [49] developed a low-input version of the MACC assay, which reduces the number of cells and improves the data quality. Local and global regional increased and inhibited accessibility and nucleosome occupancy during lymphoid development were analyzed by MACC. It was found that changes in local and global chromatin openness were consistent with the location of nucleosome occupancy and histone modification [49].

### 3.2. Nucleosome-Level 3D-Genomics Technology and Application Discovery

The genome-wide nucleosome mapping technologies of nucleosome positions, nucleosome occupancy, and chromatin accessibility mainly focus on the one-dimensional (1D) and two-dimensional (2D) levels of genome [53]. How about 3D chromatin conformation at the nucleosome level? Chromatin conformation capture technology has achieved rapid development in the last decade, Hi-C and its derivatives have spurred scientists to discover and redefine the chromatin 3D conformation. These include chromosome territories, A/B compartments, TADs, sub-TADs, and DNA-loops [16]. Due to Hi-C technology limitations, the resolution of the biochemical and molecular biological method is limited to 5 Mb~1 kb bins [54]. This does not allow us to reconstruct the 3D structure of chromatin at nucleosome-level resolution (100~200 bp) (Figure 1a) [18]. This is not to mention integrating with other omic technologies to study the impact of nucleosome structure and dynamic changes on life activities.

Interestingly, in 2015, Hsieh et al. created Micro-C technology (micrococcal nuclease chromosome conformation assay), and by using it, mapped nucleosome resolution chromosome conformation in yeast (Figure 3) [17]. Micro-C is a method that also uses micrococcal nuclease for chromatin fragmentation, which provides a high-resolution view of 3D genome structure. They proposed chromosome-interacting domains (CIDs), which are similar to but substantially shorter than the topologically associating domains (TADs) defined in mammalian cells. Besides, Micro-C found that the upstream regions of highly transcribed genes can form CID boundaries. Micro-C used formaldehyde to crosslink the nucleosome structure [19]. However, formaldehyde is a short-distance crosslinker, and some long-distance interactions are often lost. Thus, Hsieh et al. [55] identified two protein-to-protein crosslinkers that can capture the long-distance interactions easily, including disuccinimidyl glutarate (DSG, a 7.7-Å crosslinker) and ethylene glycol bis (succinimidyl succinate) (EGS, a 16.1-Å crosslinker). Based on these findings, using long crosslinkers, they immediately introduced Micro-C XL (Micrococcal nuclease-based analysis of chromosome folding using long x-linkers) technology the following year [55]. The technology provides an improved edition for Micro-C, which provides a clearer chromosome-folding atlas at mononucleosome resolution with an increased signal-to-noise ratio. They identified very robust centromere–centromere (CEN-CEN) and telomere–telomere (TEL-TEL) interactions between interphase chromosomes. In addition to this, Micro-C XL helped scientists observe increased contact frequencies across the entire gene body [54]. At that time, they did not redefine these derived new domains until the appearance of the technical article Tag-Hi-C. Tag-Hi-C literature defines this domain called gene domains [56].

Micro-C and its derivatives fill this gap by accessing the nucleosomal fiber at 100–1000 bp resolution [18]. They revealed multi-gene domains in S. cerevisiae and provide a new understanding of nucleosomal fiber models. The contemplated two-start zig-zag model for tetranucleosome organization was confirmed by Micro-C in mammalian cells [19]. After yeast, Micro-C was used to study human and mouse chromatin architecture [54]. Micro-C recapitulated the chromatin conformation defined by Hi-C, such as chromosome compartments, TADs, and DNA loops. Dots and stripes were revealed by Micro-C, which were nearly undetected in previous Hi-C studies [19]. In the 3D genome, thousands of boundaries of small (5–10 kb) interacting domains binding various transcription factors and associating with clustered promoter–enhancer interactions are regarded as “stripes”. At the end of these stripes interacting with their two anchors, “dots” are redefined at the chromatin loop anchors [19]. Moreover, strong extrusion pause sites, weaker pause sites, and extrusion-associated stripes were defined together [18]. Micro-C studies also raise some important questions. These were more clearly confirmed by subsequent studies. For example, self-interacting domains were redefined by Micro-C under loop extrusion, which was defined in a future study as a transcription reeling-in model by ChIA-Drop [68].

Benefitting from Micro-C technology, scientists have discovered many new structures. Technically, the main advantages of Micro-C and its derivatives are as follows. Firstly, Micro-C mainly measures the interaction between nucleosomes, the basic unit of chromatin, rather than the uneven size interactions of genome fragments that are interrupted and generated by interacting protein bodies after restriction enzymes digestion [2]. This enables a single nucleosome resolution, extending the study of chromatin folding to the sub-kilobase scale, which facilitated the remodeling of histone folding, E–P (enhancer–promoter), P–P (promoter–promoter) domains and chromatin folding at 30 nm [2]. Secondly, Micro-C has higher sensitivity for detecting chromatin loops. Micro-C maps can visualize more chromatin loops and are sequenced at a much lower depth than Hi-C [69]. For example, Hi-C typically requires more than 800 M–1.5 B unique reads to detect CTCF loops, while Micro-C can detect loop structures with 50–80 M unique reads, suggesting that it and its derivatives provide a cost-effective option for chromatin loop-level research [69]. Thirdly, the Micro-C XL method adopts a double cross-linking strategy, where formaldehyde and long-distance cross-linking agents (such as DSG or EGS) are used to immobilize protein–DNA, protein–protein interactions. It has been reported that formaldehyde crosslinking reacts slowly, is incomplete, and tends to reverse over time. After formaldehyde treatment, proteins remain free to diffuse. Formaldehyde crosslinking is also insufficient to generate high signal-to-noise data in yeast. The double-crosslinking strategy can help capture more complete protein–protein interactions [55]. Fourthly, Micro-C technology is easy to integrate with other omics or biochemical reaction steps. Therefore, we infer that Micro-3C, Micro-4C, Micro-5C, or Micro-ChIP will be powerful alternatives for ultra-high resolution research in the future.

### 3.3. Comparing Nucleosome-Level Omics Techniques with Other Genomics Techniques

In addition to using Mnase digestion step as the key to study the chromatin landscape at the nucleosome level, classical epigenetic techniques DNase-Seq [61], ATAC-Seq [62], ChIP-Seq [59], ChIP-exo [60], and NOMe-Seq [64], etc. have also attempted to increase the resolution to study nucleosome dynamics (Table 1).

Deoxyribonuclease I (DNase I) is an enzyme that randomly cuts DNA at every site. Along with DNase I digestion and high-throughput sequencing, DNase I hypersensitive site sequencing (DNase-seq) was developed [72]. In the 1960s and 1970s, deoxyribonuclease I (DNase I) was used to probe the structure of nucleosomes, but currrently in the NGS period, it is mainly used to study the NDRs. Zhong et al. found that DNase I could be used to precisely map the position of the whole genome of the nucleosome in vivo, and demonstrated that DNA-based methods can be easily applied to any organism by generating maps in yeast and humans [73]. Coincidentally, chromatin immunoprecipitation sequencing (ChIP-seq) is a technique for genome-wide analysis of DNA-binding proteins, histone modifications, or nucleosomes [59]. Using this technique, the distribution of binding sites containing specific histone modifications could be mapped across the genome. However, ChIP-Seq has the disadvantages of low resolution, data quality dependent on antibody quality, time-consuming and laborious antibody screening, and high cost. Apart from this, since nucleosomes are typically positioned using nuclease digestion, Kelly et al. [42] developed a method based on the combined action of DNA methylation and nucleosome positioning, Nucleosome Occupancy and Methylome-sequencing (NOMe-seq) technology, which uses GpC methyltransferase (M.CviPI) and NGS to generate a high-resolution nucleosome genome-wide mapping footprint [42]. Rhie et al. [74] used NOMe-seq to find that in four cancer cell lines (kidney, colon, prostate, and breast cancer), the transcription factor ZFX associated with tumorigenesis and proliferation is in the open chromatin region of the TSS and the first downstream (+1) nucleosome. This suggests that ZFX may play a key role in the promoter structure, and may provide new insights into the regulation of transcription, chromatin structure, and cancer transcriptomes [74]. Besides, as the roles of nucleosome occupancy and epigenetic reprogramming of distal regulatory elements in cancer are unclear, Taberlay et al. [75] used the simultaneous genome-wide mapping of NOMe-seq to assess the overall range of epigenetic alterations in the enhancer and insulator elements in prostate and breast cancer cells. The study also showed that the failure of NDRs and TFs of distal regulatory elements in cancer cells was accompanied by susceptibility to DNA hypermethylation of enhancer and insulator elements, which may contribute to changes in genome structure and epigenetic disorders [75]. However, these techniques have limited resolution compared to Mnase-seq, Micro-C, and their derivatives, but provided very rich nucleosome-level research content at an early stage.

Interestingly, other omics or biochemical reaction steps can be integrated to develop the brand new Micro-C-based technologies, which would be the trends in the future nucleosome level 3D genomic research. Hua et al. [76] developed a high-resolution Mico-C method (Micro-Capture-C, MCC) that allows physical contacts at the nucleosome level or nearly base-pair resolution between specific regulatory elements to be determined to obtain insights into gene regulation (Figure 4). They found that the loop extrusion happened at active promoters and enhancers dependent on cohesin loading. Then, they form tissue-specific chromatin domains without changes in CTCF binding [76]. The method integrates Hi-C, Micro-C, Promoter Capture Hi-C, 4C-seq, and Next Generation (NG) Capture-C, enabling physical contacts between different types of regulatory elements at base-pair resolution. Compared with Promoter-capture Hi-C (PCHi-C), 4C-seq, and NG capture-C, MCC has greatly improved resolution at the nucleosome level [76]. MCC draws on the idea of the PCHi-C, which could be regarded as a further upgrade of Micro-C.

There is another interesting piece of research work. Masae et al. [77] established a new nucleosome-level approach to reveal the 3D chromatin spatial distribution and genome-wide localization of nucleosomes on the genome, termed Hi-CO. This approach combines higher nucleosome-resolution techniques with annealing-molecular dynamics (SA-MD) simulations to reveal distinct nucleosome folding motifs spanning the yeast genome. They proposed two basic secondary structure motifs in nucleosome folding: α-tetrahedron and β-rhombus, and used mutants, and cell cycle-synchronized cells to further reveal nucleosome-specific localization and orientation [77]. This study proposes the organizing principles of nucleosome folding. For methodology, Hi-CO adds a specific linker that can identify the direction when the histone-wrapped DNA fragments are proximity-ligated. Hi-CO beautifully shows the positioning and folding orientation of the nucleosomes. In particular, the wrapping direction of DNA was captured by Hi-CO for the first time. Hi-CO draws on the idea of adding a linker to distinguish noise in ChIA-PET, and converts it into the direction of “in and out” of DNA-wrapped nucleosomes, which could also be regarded as a continued upgrade of Micro-C.

With the development and application of omics technology, the supporting bioinformatics analysis algorithms, software, and pipelines have also been gradually coded.

## 4. Research and Development Progress on Bioinformatics Analysis Tools and Pipelines for Nucleosomics

Nucleosome positioning is an important piece for studying chromatin. The developing high-throughput sequencing further provides a way to analyze the genome-wide scale nucleosome positioning across different species and data types. A high resolution of nucleosome positions requires up to billions of reads order, which means the storage space will be terabytes [27]. To collect these extensive sequencing experiment data as well as to track the corresponding publications, a complete list of databases were created [78]. Databases from large international organizations like ENCODE and FANTOM provide gold standard and sharing of high-quality referable data for subsequent related studies [79,80]. However, the data on gene regulation produced by various laboratories were lacking. These indispensable data are present in GEO and SRA repositories but still lack uniform annotation and processing [81], and the search process is not convenient. To solve these problems, convenient for large-scale integrated analysis, the development and application of standardized workflows and databases are motivated by efficient usage of the vast but unsystematic published data. The databases are listed in Table 2, including various sequencing data for different species.

Nucleosome positioning depends on cell type and state mainly. As another dataset of NGS analysis, the first steps for read mapping and quality control of nucleosome positioning sequencing data require standard NGS software [78]. Diverging from the CHIP-seq experimental data, millions of small, fuzzy, and specified enrichment peaks needed to be dealt with in the nucleosome positioning research. Teif summed up over 20 computational methods about the principal analysis of nucleosome positioning sequencing data according to five preferences, e.g., detect enriched peaks around 147 bp or the continuous nucleosome occupancy profile instead, operates with Monte Carlo simulations, taking advantage of some available nucleosome and generic genomic visualizing tools or browsers [78]. With the rise of interdisciplinary disciplines, analytical tools for nucleosomes are constantly being created, including developing the deep learning model [82], integrating various sequencing data, model, characteristics of nucleosome organization, probabilistic methods, and even dynamics occupancy about chromatin [83,84,85]. These tools are also convenient to find, use, or even modify with the development of algorithmic technology, code visualization, and disclosure. The description and use of these developed algorithms, language, and URL are displayed in Table 3.

## 5. Conclusions and Future Perspectives of Nucleosome-Level Genomics Technologies and Applications

In this paper, we reviewed the scientific discovery, development, and related research of nucleosomes, described the significance of studying chromatin nucleosomes and nucleosome-level DNA, then summarized genomics techniques and progress for studying epigenetic phenomena at the nucleosome level, including genomics and 3D-genomics technologies. The bioinformatics analysis tools and databases were also discussed in the review. We expect this review could enable readers to understand the field of nucleosome-omics, which would open a new perspective on the epigenetic code and 3D genome landscape.

Finally, the future perspectives of nucleosome-level genomics technologies and their improvement directions are considered here. Towards scientific experiments, on the one hand, the combination of chromatin capture technology and nucleosome positioning technology helps to analyze the 3D interactions of nucleosomes at a more refined chromatin level and can better understand the relationship between chromatin structure and its function. Learning from the idea of the SPRITE technology [16], the multi-way nucleosome-level interactions could be captured by adding multi-adapters to the mononucleosome interactive complex for replacing proximity-ligation. Additionally, ChIA-Drop can also provide a design such that we could pass the mononucleosome interactive complex through microfluidics and develop the nucleosome-level ChIA-Drop [16]. Intriguingly, some non-eukaryotic organisms (such as viruses) with relatively small genomes also have nucleosomes [95]. The study of the structural characteristics and their biological roles of viral nucleosomes, whether in terms of chromatin assembly or evolution, is of great significance. In bioinformatics analysis, the first point is to develop multi-information fusion algorithms to accurately locate the positions of various eukaryotic nucleosomes; the second point is to integrate multiple types of data to locate nucleosomes. We should strengthen the development of algorithms for theoretical prediction. The third point, since it is difficult to distinguish the vaguely positioned special nucleosomes from the well-positioned common nucleosomes that exist simultaneously in the chromosomal region, it is necessary to reconstruct a more accurate nucleosome distribution model through the development of improved algorithms or new algorithms in the future.

## Figures and Tables

**Figure 1 genes-13-01114-f001:**
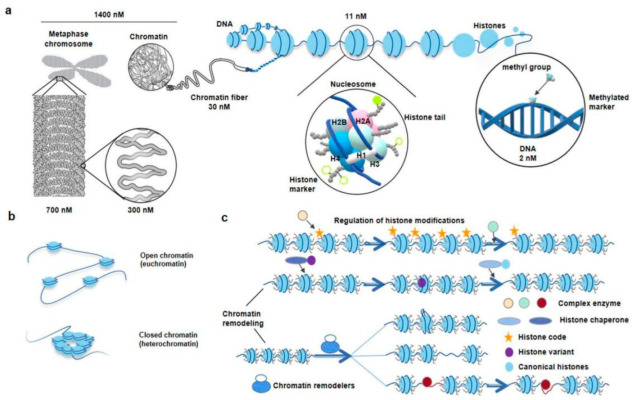
The diagram of hierarchical structures from chromatin to nucleosome in eukaryotes. (**a**), Zoom in chromosome or chromatin to DNA. The DNA double helix wraps around histone octamers to form nucleosomes, the basic structural units of chromatin. The classic histone octamer is made up of four types of histone (H2A/H2B/H3/H4) and has eight histone tails. Covalent modification markers on histone tails play an important role in regulating the chromatin structure and function. DNA methylation markers are also epigenetic codes. (**b**), Different states of chromatin. The regions of chromatin, called euchromatin, exist in an extended and loosely packed state, which is conducive to transcription. Conversely, densely packed heterochromatin is silent. (**c**), Regulation of histone modifications and chromatin remodeling. Recognition, reading, and removal of histone modifications depend on the Writer, Reader, and Eraser. The complex enzymes responsible for the alteration of histone modifications often have more than one ability. When the reading modules of these complexes bind to the corresponding sites, their writing or erasure modules are activated and work at nearby sites. ATP-driven chromatin remodelers, which are complex of multiple subunits, catalyze chromatin remodeling. Meanwhile, the nucleosome structure usually changes in four ways: replacement, dissociation, removing, and slide. Replacement indicates chromatin remodelers, which catalyze replacement between canonical histone and histone variants in nucleosomes. Dissociation indicates that the double helix DNA wrapped around the nucleosome loosens. Removing indicates the disintegration of nucleosomes. Slide indicates chromatin remodelers can allow nucleosomes to slide along DNA without unwinding the DNA double strand [32]. Courtesy: National Human Genome Research Institute (https://www.genome.gov/genetics-glossary/Chromatin, accessed on 19 May 2022; https://www.genome.gov/genetics-glossary/Nucleosome, accessed on 19 May 2022).

**Figure 2 genes-13-01114-f002:**
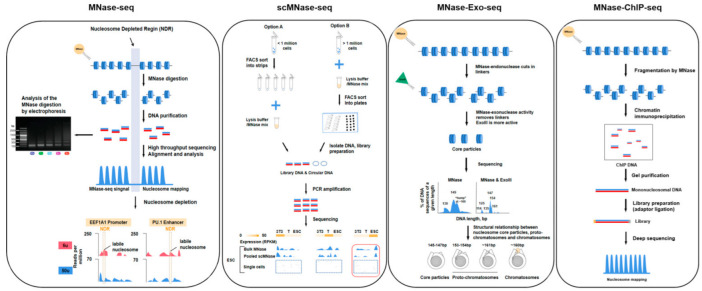
The methodology and brief analysis result overview of MNase-seq and its derivatives. Schematic view of the MNase-seq, scMNase-seq, MNase-Exo-seq, and MNase-ChIP-seq procedures and principles. All these technologies use MNase to digest internucleosomal DNA and allow nucleosome DNA to be released. Subsequent steps for MNase-seq include digestion, DNA purification, library preparation, and high-throughput sequencing. The high-resolution location of the mononucleosome and location of the NDRs can be determined. MNase-seq panel was adapted with permission from Ref. [50]. 2021, Springer Nature. scMNase-seq adds the step of cell sorting on the basis of MNase-seq. As shown in the Figure, two methods of sorting and digestion are provided. scMNase-seq panel was adapted with permission from Refs. [40,51]. 2019, 2018, Springer Nature. MNase-Exo-seq increases the ExoIII digestion step, leading to significant spikes in core particle size and a more accurate depiction of nucleosome location. MNase-Exo-seq panel was adapted from Ref. [12] with Open Access and no copyright issue. MNase-ChIP-seq increases the chromatin immunoprecipitation step after digestion and can perform the amplification of specific mononucleosomal DNA [52]. MNase-ChIP-seq panel was adapted with permission from Ref. [13]. 2011, Elsevier.

**Figure 3 genes-13-01114-f003:**
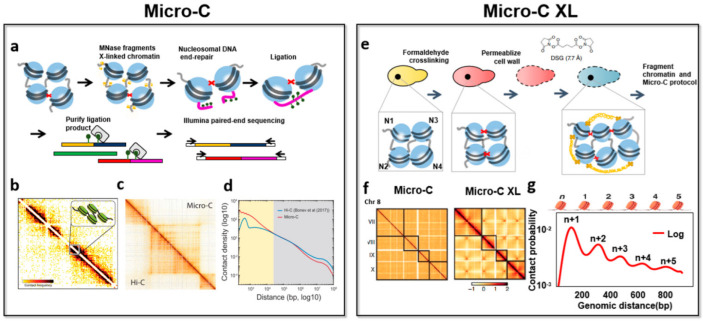
The methodology of Micro-C and Micro-C XL. (**a**), Outline of key steps in the Micro-C protocol. Adapted with permission from Ref. [17]. 2015, Elsevier. (**b**), Single−nucleosome resolution contact matrix. Reprinted with permission from Ref. [15]. 2015, Elsevier. (**c**), Heatmap of Micro-C−specific dots and stripes for loop interactions in HFFs in chromosome 8. Corresponding Micro-C and Hi−C heatmaps are shown above and below the diagonal lines, respectively. Reprinted with permission from Ref. [69]. 2020, Elsevier. (**d**), Difference plot of interaction decaying rates of Micro-C and Hi−C. X axis and Y axis represent distances of 100 bp to 10 MB between contact loci and contact density normalized by sequencing depth, respectively. Reprinted with permission from Ref. [70]. 2020, Elsevier. (**e**), An overview of the Micro-C XL protocol. Adapted with permission from Ref. [55]. 2016, Springer Nature. (**f**), Interaction heatmaps of Micro-C data and Micro-C XL from yeast chromosomes VII to X in 10 kb bin resolution. Adapted with permission from Ref. [55]. 2016, Springer Nature. (**g**), The decaying line of contact probability with genomic distance of Micro-C XL data is shown in 1 kb bin resolution. Adapted from Ref. [71] with Open Access and no copyright issue.

**Figure 4 genes-13-01114-f004:**
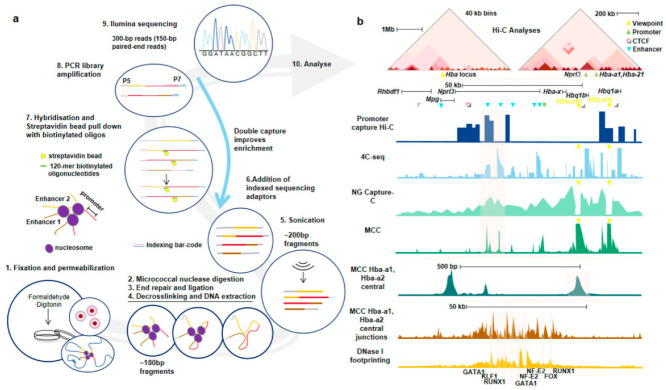
MCC, an upgraded version of Micro-C. (**a**), Library construction process of MCC. Like other 3C technologies, MCC captures interactions through chemical cross-linking. MCC uses micrococcal nuclease (MNase) which is independent of DNA sequence as the DNA molecular scissor to achieve random fragmentation, which is the key step to improve resolution. The Oligonucleotide probe tends to bind to the center of the hypersensitive site. Hybridization with probes is conducted to enrich the specific interactions. (**b**), MCC has a higher resolution and can capture interactions that the other 3C technologies do not. Adapted with permission from Ref. [76]. 2021, Springer Nature.

**Table 1 genes-13-01114-t001:** Summary of different experimental techniques for nucleosomics.

Technology	Sequence Type	Start Material	Enzyme Digestion	Advantage	Feature	Reference
MNase-seq	Single-end or paired-end sequencing	10–20 million cells	MNase	The whole genome can be measured, high resolution, low technical difficulty.	MNase has sequence bias, it cuts DNA upstream of A or T faster than upstream of G or C, traditional method requires a large number of cells, easy to cause technical error.	[57,58]
scMNase-seq	Single-end or paired-end sequencing	>0.1 million cells	MNase	Provides a method for determining nucleosome localization and chromatin accessibility in single-cell or low-input materials, few cells are required, high resolution, low technical difficulty.	Low capture rate, information may be missing, lower throughput.	[40]
ULI-MNase-seq	Paired-end sequencing	10–15 pronuclei per replicate	MNase	Low cell initiation.	It requires extremely high proficiency and skill. It easily loses cell and library DNA.	[41]
MNase-Exo-seq	Paired-end sequencing	10–20 million cells	MNase, exonuclease III	The perfectly clipped core particle has a major sharp peak corresponding to it, weaker signals that cannot be detected in MNase-seq data are evident.	Detailed analysis of nucleosome location is complicated, it is possible to misestimate the occupancy of a specific nucleosome positions.	[12]
ChIP-seq	Single-end or paired-end sequencing	>1 million cells	Priority with MNase	High resolution and low noise, high genome coverage and wider dynamic range, the method is more mature and widely applicable.	Data quality depends on antibody quality, and antibody screening is time-consuming and costly.	[58,59]
ChIP-exo	Single-end or paired-end sequencing	>1 million cells	Exonuclease	High resolution, defines genomic binding locations, more precisely determine the location of protein gene interactions in the genome, few false positives or negatives Binding-site complexity.	Multiple binding of a single protein cannot be detected.	[60]
ChIP-MNase	Single-end or paired-end sequencing	10–20 million cells	MNase	High resolution, nucleosome localization analysis in specific position of the genome and differential analysis of alleles undergoing different molecular processes.	Need for precise selection of antibodies, antibody repertoire may be incomplete, Other features similar to MNase-seq.	[61]
ATAC-seq	Paired-end sequencing	500–50,000 cells	Tn5 transposase	Simple method, short experiment period, few cells are required, high resolution and good repeatability.	Conventional data analysis methods have limitations, Tn5 transposase is expensive.	[58,62]
DNase-seq	Single-end or paired-end sequencing	>1 million cells	DNase I	Simple method, high resolution, the most active regulatory regions can be identified from many cell types.	Traditional method requires a large number of cells, precise control of enzyme quantity, time consuming; it was not easy to determine the precise activity and function which were associated with each regulatory region.	[58,63]
NOMe-seq	Paired-end sequencing	Need to test	GpC methyltransferase	Does not depend on knowing the exact modification of surrounding nucleosomes. It can provide localization information of multiple nucleosomes on both sides of each open regulatory element, nucleosome localization and DNA methylation degree can be analyzed simultaneously.	Requires a large number of cells and data analysis is difficult.	[58,64]
Micro-C	Paired-end sequencing	0.001–5 million cells	MNase	The signal-to-noise is improved, high resolution, reveals the chromosome folding of nucleosome resolution.	The observed chromosome structure will be biased, difficulty recovering known higher-order interactions.	[17]
Micro-C XL	Paired-end sequencing	1 million cells	MNase	The signal-to-noise is improved, improved structure visualization, chromosome folding can be detected from nucleosomes to whole genomes, adds some subtle details to the Micro-C map.	Requires one more step of cross-linking, it may take many attempts to find the best conditions.	[55]
MACC-seq	Single-end or paired-end sequencing	1 million cells per reaction	MNase	Profiles both open and closed genomic loci simultaneously, combined with ChIP specificity to enrich histone modification-associated DNA fragments.	Traditional method requires a large number of cells.	[65]
MH-seq	Paired-end sequencing	10–20 million cells	MNase	Simple procedures, enables detection of distinct types of open chromatin.	Traditional method requires more cells, it is not easy in plants to establish Single-cell-based MH-seq, application in plants has limitations, high requirements for nuclear quality.	[66]
Array-seq	Paired-end sequencing	10–20 million cells	MNase	Reveals linker length and array regularity in unmappable areas.	Traditional method requires more cells, titration test required.	[44]
MRE-seq	Paired-end sequencing	Need to test	Methylation-sensitive restriction enzymes	The methylation status of most repeats can be revealed, the methylation state of a local region or a single CPG can be addressed, MREs are inexpensive.	The recognition range of methylation events is limited, and only those within MRE recognition sites can be detected.	[67]

**Table 2 genes-13-01114-t002:** Nucleosome prediction database.

Database	Description	Data Type	Species	Source	Reference
GTRD (Gene Transcription Regulation Database)	The largest integrated resource of data on transcription regulation in eukaryotes, which contains uniformly annotated and processed NGS data, the results of the meta-analysis, and the sets of non-redundant and reproducible TFBSs for each TF.	ChIP-seq, ChIP-exo, DNase-seq, MNase-seq, ATAC-seq, RNA-seq	*Homo sapiens*, *Mus musculus*, *Rattus norvegicus*, *Danio rerio*, *Caenorhabditis elegans*, *Drosophila melanogaster*, *Saccharomyces cerevisiae*, *Schizosaccharomyces pombe*, *Arabidopsis thaliana*	http://gtrd.biouml.org/ (accessed on 8 January 2021)	[81]
NPRD (Nucleosome Positioning Region Database)	It is compiling the available experimental data on locations and characteristics of nucleosome formation sites (NFSs), and is the first curated NFS-oriented database.	The type used in original paper	*S. cerevisiae*, *Homo sapiens*, *Simian virus*, *Mus musculus*, *Drosophila melanogaster*, *Rattus norvegicus*, *Tetrahymena thermophila*, *Chironomus tentans*, *Mouse mammary tumor virus*, *Xenopus laevis* (African clawed frog), Oryctolagus cuniculus (rabbit)	http://srs6.bionet.nsc.ru/srs6/ (accessed on 20 June 2022)	[86]
ChIP-Atlas	An integrative, comprehensive database to explore public Epigenetic dataset, covers almost all public data archived in Sequence Read Archive of NCBI, EBI, and DDBJ with over 224,000 experiments.	ChIP-Seq, DNase-Seq, ATAC-Seq, Bisulfite-Seq	*H. sapiens*, *M. musculus*, *R. norvegicus*, *D. melanogaster*, *C. elegans*, *S. cerevisiae*	https://chip-atlas.org/ (accessed on 4 January 2021)	[87]
CistromeDB	A resource of human and mouse cis-regulatory information, which map the genome-wide locations of transcription factor binding sites, histone post-translational modifications, and regions of chromatin accessible to endonuclease activity.	ChIP-seq, DNase-seq, ATAC-seq	*H. sapiens*, *M. musculus*	http://cistrome.org/db/#/ (accessed on 20 June 2021)	[88]
ENCODE	Integrative-level annotations integrate multiple types of experimental data and ground level annotations. Ground-level annotations are derived directly from the experimental data, typically produced by uniform processing pipelines.	ChIP-seq, DNase-seq, ATAC-seq, TFChIP-seq, RNA-seq, eCLIP-seq, ChIA-PET, Hi-C, RRBS, WGBS, RAMPAGE	*H. sapiens*, *M. musculus*, *D. melanogaster*, *C. elegans*	https://www.encodeproject.org (accessed on 30 January 2019)	[89]
ChIPBase	Decoding the encyclopedia of transcriptional regulations of ncRNAs and PCGs.	ChIP-seq, ChIP-exo, ChIP-nexus, MNChIP-seq	*H. sapiens*, *M. musculus*, *R. norvegicus*, *D. rerio*, *X. tropicalis*, *C. elegans*, *D. melanogaster*, *S. cerevisiae*, *A. thaliana*, *G. gallus*	https://rna.sysu.edu.cn/chipbase3/index.php (accessed on 1 July 2021)	[90]
ReMap 2020 3rd release	Information of regulatory regions from an integrative analysis of Human and Arabidopsis DNA-binding sequencing experiments.	ChIP-seq, ChIP-exo, ChIP-nexus, DAP-seq	*H. sapiens*, *A. thaliana*	http://remap.univ-amu.fr (accessed on 7 January 2022)	[91]
Factorbook	A transcription factor (TF)-centric repository of all ENCODE ChIP-seq datasets on TF-binding regions, as well as the rich analysis results of these data.	ChIP-seq	*H. sapiens*, *M. musculus*	http://factorbook.org (accessed on 20 June 2022)	[92]
NucMap	Genome-wide nucleosome positioning map across different species.	MNase-Seq	*Arabidopsis thaliana*, Caenorhabditis elegans, Candida albicans, Danio rerio, Drosophila melanogaster, Homo sapiens, Mus musculus, Neurospora crassa, Oryza sativa, Plasmodium falciparum, Saccharomyces cerevisiae, Schizosaccharomyces pombe, Trypanosoma brucei, Xenopus laevis, Zea mays	http://bigd.big.ac.cn/nucmap (accessed on 4 April 2022)	[93]
NucPosDB	Database reporting experimental nucleosome maps in vivo across different cell types and conditions, cell-free DNA (cfDNA) datasets in people and model organisms, processed stable-nucleosome regions, as well as software for computational analysis and modeling of nucleosome positioning and “nucleosomics” analysis for medical diagnostics.	MNase-seq, ChIP-seq, MH-seq, MPE-seq, MiSeq, NOME-seq, RED-seq, Nanopore-seq, Fiber-seq, Ucleosome-scale mapping of 3D genome contact, Micro-C	*S. cerevisiae*, *M. musculus*, *H. sapiens*, *D. melanogaster*, *A. thaliana*, *S. pombe*, *C. elegans*, *N. crassa*	https://generegulation.org/nucposdb/ (accessed on 20 June 2022)	[94]

**Table 3 genes-13-01114-t003:** Recently developed computational tools for the analysis of experimental nucleosome data.

Algorithm	Web Server/GUI	Feature	Input Dataset	Languages	Source	Reference
CAESAR	+/−	Connecting epigenomics and chromatin organization at the nucleosome resolution.	Epigenomic features and Hi-C contact maps	Python	https://github.com/liu-bioinfo-lab/caesar (accessed on 1 March 2022)	[82]
Factor-agnostic chromatin occupancy profiles from MNase	+/−	Links changes in chromatin at nucleotide resolution with transcriptional regulation.	MNase-seq and RNA-seq data	Python, Shell	https://github.com/HarteminkLab/cadmium-paper (accessed on 18 April 2021)	[85]
NucHMM	+/−	Identifies functional nucleosome states associated with cell type-specific combinatorial histone marks and nucleosome organization features.	MNase-seq and ChIP-seq data	Python, C++, Makefile	https://github.com/KunFang93/NucHMM (accessed on 2 June 2022)	[83]
ProbC	+/−	Decomposes Hi-C and Micro-C interactions by known chromatin marks at genome and chromosome levels.	Hi-C and Micro-C data	Python	http://www.github.com/seferlab/probc (accessed on 19 March 2022)	[84]

Footer: +:Yes; −:No.

## Data Availability

Not applicable.

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
