# Peer review of "Nucleosome-Omics: A Perspective on the Epigenetic Code and 3D Genome Landscape"

_genes, 2022, doi:10.3390/genes13071114_

Round 1

Reviewer 1 Report

The submitted manuscript entitled ‘Nucleosome-omics: a perspective on the epigenetic code and 3D genome landscape overview the relation between in respect to epigenetic information.’ represents an overview of the impact of nucleosome as basal chromatin structure on the regulation of epigenetic code and the technology that help to understand the landscape, which in turn define the transcription.

The text is well written and easy to understand however, sometimes contains ‘non-orthodox’ scientific language e.g. Lines 73-74 ’….we could observe some good chromosome packages called metaphase chromosome….’.

All figures have very low resolution. In figure embedded text is not readable. Figure 1 legend demand extended description of depicted schemes.

In the introduction, section lines 91-102 describing relationship of epigenetic and transcriptional regulation with factors of preinitiation complex, Pol2 and histone modifications should briefly mention nuclear PIPs. More specifically, the nuclear PIPs are critical interactors and regulators of chromatin remodeling and transcription and thus are crucial for establishment of transcriptionally competent landscape of the cell nucleus (described in recent papers PMID: 33406800 and PMID: 33513445).

Author Response

Reviewer 1:

  1. The reviewer’s comment: The text is well written and easy to understand however, sometimes contains ‘non-orthodox’ scientific language e.g. Lines 73-74 ’….we could observe some good chromosome packages called metaphase chromosome….’.

The authors answer: We thank the reviewer for his/her affirmation of our work. According to the comment, we have revised the sentence to “we could observe some regular-shaped and compact chromosome packages called metaphase chromosomes, and sometimes, loose and seemingly irregular chromatin in other interphase cells[20].” The new sentence is at Lines 75–77 in the revision.

  1. The reviewer’s comment: All figures have very low resolution. In figure embedded text is not readable. Figure 1 legend demand extended description of depicted schemes.

The authors answer: We are thanked for the review’s valuable comment. We have replaced the figures with the high-resolution ones. We also checked all the Figure legends. Specifically,  the legend in Figure 1 has been revised to “Figure 1. The diagram of hierarchical structures from chromatin to nucleosome in eukaryotes. (a), Zoom in chromosome or chromatin to DNA. The DNA double helix wraps around histone octamers to form nucleosomes, the basic structural units of chromatin. The classic histone octamer is made up of four types of histone (H2A/H2B/H3/H4) and has eight histone tails. Covalent modification markers on histone tails play an important role in regulating the chromatin structure and function. DNA methylation markers are also epigenetic codes. (b), Different states of chromatin. The regions of chromatin, called euchromatin, exist in an extended and loosely packed state, which is conducive to transcription. Conversely, densely packed heterochromatin is silent. (c), Regulation of histone modifications and chromatin remodeling. Recognition, reading, and removal of histone modifications depend on the Writer, Reader, and Eraser. The complex enzymes responsible for the alteration of histone modifications often have more than one ability. When the reading modules of these complexes bind to the corresponding sites, their writing or erasure modules are activated and work at nearby sites. ATP-driven chromatin remodelers, which are complex of multiple subunits, catalyze chromatin remodeling. Meanwhile, the nucleosome structure usually changes in four ways: replacement, dissociation, removing and slide. Replacement indicates that chromatin remodelers, which catalyze replacement between canonical histone and histone variants in nucleosomes. Dissociation indicates that the double helix DNA wrapped around the nucleosome loosens. Removing indicates the disintegration of nucleosomes. Slide indicates chromatin remodelers can allow nucleosomes to slide along DNA without unwinding the DNA double strand[34].” at Lines 119–134 in the revision.

  1. The reviewer’s comment: In the introduction, section lines 91-102 describing relationship of epigenetic and transcriptional regulation with factors of preinitiation complex, Pol2 and histone modifications should briefly mention nuclear PIPs. More specifically, the nuclear PIPs are critical interactors and regulators of chromatin remodeling and transcription and thus are crucial for establishment of transcriptionally competent landscape of the cell nucleus (described in recent papers PMID: 33406800 and PMID: 33513445).

The authors answer: We thank the reviewer for the valuable comment. It is an important point. In the revision, we have eluded the point in. The reviewer can find the sentence “Besides, nuclear phosphatidylinositol phosphates (PIPs) are also critical interactors and regulators of chromatin remodeling and transcription and thus are crucial for establishing a transcriptionally competent landscape of the cell nucleus[31,32].” in lines 105–108.

Reviewer 2 Report

Kong et al summarized current progress in nucleosome-omics technologies for studying genome organization and 3D genome landscape in the review. This review is of great importance for researchers interested in utilizing nucleosome-omics technologies. While the work nicely  compared different types of technologies that can resolve genome/3D genome organization and epigenetic modifications at the nucleosome-level, here are a few comments to improve the manuscript.

1). While the authors nicely summarized and compared different technologies, the applications in biological researches are not be sufficiently described. It will be helpful if the authors can used more examples to illustrate what kinds of scientific discoveries have been made using these technologies.

2). The current figures are quite blurr to show details/texts, which need to be improved. 

3) The manuscript suffers from many grammar problems all over the place, which need to be thoroughly checked. 

4) Line 385-386, the description and use of these algorithms, language and url are displayed in the Table 3 rather than in Table 1.

Author Response

Reviewer 2:

  1. The reviewer’s comment: While the authors nicely summarized and compared different technologies, the applications in biological researches are not be sufficiently described. It will be helpful if the authors can used more examples to illustrate what kinds of scientific discoveries have been made using these technologies.

The authors answer: We thank the reviewer for the valuable comment. Readers should find this information in our manuscript. Therefore, according to the reviewer’s comment, we have described in detail the application of the different techniques in biological research: ”Furthermore, we illustrate some kinds of scientific discoveries have been made using these MNase-seq derivate technologies. Single-cell micrococcal nuclease sequencing (scMNase-seq) was explored to detects genome-wide nucleosome orientation and chromatin accessibility from single cells or a small number of cells by Gao et al. Compared with MNase-seq, the advantages of scMNase-seq is that can become a single-cell suspension from dissociated tissues. It is significant to reveal epigenetic heterogeneity during the period of normal tissue development or pathological processes and reveal important information for highlighting the molecular mechanisms[43]. To illustrate the new mechanisms of chromatin remodeling, Wang et al. used ultra low-input MNase-seq to analyze the dynamic changes of genome-wide nucleosome occupancy and positioning of mouse embryos during the first 12 h after fertilization. They revealed some novel molecular regulation mechanisms of the ZGA process, including the detailed pattern of NDR rebuilding in promoters and nucleosome positioning dynamics in TF binding motifs in mice[44]. Based on MNase-seq, Ocampo et al. introduced E. coli exonuclease III (Exo III) and MNase in simultaneous digestion of chromatin and reconstituted nucleosomes in budding yeast nuclei. They discovered two new intermediate particles that formed when it removes histone H1. They showed the difference among core particles, proto-chromatosomes and chromatosomes (Fig 2, MNase-Exo-seq)[12]. By analyzing the histone content of all MNase-sensitive complexes using MNase-ChIP-seq, researchers found that yeast promoters are predominantly bound by non-histone protein complexes and MNase sensitivity does not indicate instability, but rather the preference of MNase for A/T-rich DNA compared with G/C-rich nucleosomes[51]. Since MNase could recognize more open chromatin regions than DNase and Tn5, Zhao et al. developed a technique called MNase hypersensitivity sequencing (MH-seq), which was used to show the open chromatin regions in Arabidopsis thaliana. Compared with DNase-seq or ATAC-seq, the genome of open chromatin regions identified by this method has a wider range. This provides a new method for the recognition and classification of open chromatin in plants and animals[48]. Array-seq is another technology, including some steps of electrophoresing the DNA digested by MNase on the Bioanalyzer equipment, and selecting the samples with the appropriate degree of enzyme digestion to sequence on the Oxford Nanopore Minion sequencer. Using this method, Baldi et al. determined array regularity and linker length throughout the Drosophila genome, even in non-localizable regions, and revealed an inverse correlation of regularity with transcriptional activity[49]. ChIP-exo is the high precision version of ChIP-seq. It was applied to map the accurate DNA fragments bound by specific proteins in mammalian cell lines. ChIP-exo was used to reveal the genome-wide binding sites of GATA1 and TAL1 transcription factors in mouse erythroid cells. They found that TAL1 is directly recruited to DNA by protein-protein interactions with GATA1 during erythroid  differentiation[52,53].” in lines 192–231 and “Traditional MACC-seq requires a large number of cells, and Lion et al. developed a low-input version of the MACC assay, which reduces the number of cells and improves the data quality. Local and global regional increased and inhibited accessibility and nucleosome occupancy during lymphoid development were analyzed by MACC. It was found that changes in local and global chromatin openness were consistent with the location of nucleosome occupancy and histone modification[54]” in lines 238-243.

  1. The reviewer’s comment: The current figures are quite blurr to show details/texts, which need to be improved. 

The authors answer: We thank the reviewer for the valuable comment. We replaced all figures with high-resolution ones. And we also reviewed all the legends to make sure the text was readable for readers. 

  1. The reviewer’s comment: The manuscript suffers from many grammar problems all over the place, which need to be thoroughly checked. 

The authors answer: We thank the reviewer for the valuable comment. In the revision, we have asked Dr. Jun Yin from NIH/NINDS to help us revise the manuscript. The grammar has been comprehensively improved in the revision.

  1. The reviewer’s comment: Line 385-386, the description and use of these algorithms, language and url are displayed in the Table 3 rather than in Table 1.

The authors answer: We thank the reviewer for the valuable comment. This is an important point. We have modified the description and use of these algorithms, language and URL in lines 451-452 to Table 3.
